# Spectral Dependence of the Photoplastic Effect in CdZnTe and CdZnTeSe

**DOI:** 10.3390/ma14061465

**Published:** 2021-03-17

**Authors:** Jan Franc, Václav Dědič, Pavel Moravec, Martin Rejhon, Roman Grill, Hassan Elhadidy, Vladimír Šíma, Miroslav Cieslar, Maroš Bratko, Utpal Roy, Ralph B. James

**Affiliations:** 1Faculty of Mathematics and Physics, Charles University, Ke Karlovu 5, CZ-121 16 Prague, Czech Republic; dedicv@karlov.mff.cuni.cz (V.D.); moravec@karlov.mff.cuni.cz (P.M.); Martin.Rejhon@mff.cuni.cz (M.R.); grill@karlov.mff.cuni.cz (R.G.); Hassan.Elhadidy@mff.cuni.cz (H.E.); Vladimir.Sima@mff.cuni.cz (V.Š.); Miroslav.Cieslar@mff.cuni.cz (M.C.); Maros.Bratko@mff.cuni.cz (M.B.); 2Physics Department, Faculty of Science, Mansoura University, Mansoura 35516, Egypt; 3Savannah River National Laboratory, Savannah River Site, Aiken, SC 29808, USA; Utpal.Roy@srnl.doe.gov (U.R.); ralph.james@srnl.doe.gov (R.B.J.)

**Keywords:** photoplasticity, CdZnTe, CdZnTeSe, spectral dependence

## Abstract

We studied the spectral dependence of the Vickers microhardness HV0.025 of CdZnTe and CdZnTeSe samples upon illumination and found out that it increases over the entire applied spectral range of 1540–750 nm. We also found out that the photoconductivity and microhardness are correlated. We observed changes in the correlation diagram (change of slope and an abrupt change of HV0.025 at several wavelengths of the illuminating light). Based on measurements of the relative changes of the space charge upon illumination using the Pockels effect, we suggest that the observed spectral dependence of positive photoplastic effect in CdZnTe and CdZnTeSe can be explained by the trapping of photoinduced electrons and holes, which affects the motion of the partial dislocations. The underlying physical explanation relies on the assumption that reconstructed bonds break before dislocation glide.

## 1. Introduction

CdTe and Cd_1-x_Zn_x_Te (x = 0.1–0.2) are widely used II-VI semiconductors applied in solar cells [1], room-temperature X-ray and gamma-ray detectors [2], substrates for the narrow gap (HgCd)Te epitaxial infrared detectors [3], electro-optical modulators and other optical applications [4]. The crystal growth, as well as the processing of the as-grown material, is principally affected by the mechanical characteristics of the materials. CdTe has a rather low hardness (Vickers hardness was reported in the range of 50–60 kg/mm^2^ [5,6]). Therefore, the mechanical treatment and processing of CdTe are complicated by the risk of the generation of extensive defects that can substantially influence its electronic and optical properties. As-grown crystals are often decorated by twins or cracks [2] induced by mechanical stress forced by dissimilar thermal expansion coefficients of the CdTe and wall of the growth ampoule.

The so-called photoplastic effect (PPE) expresses the influence of illumination on plasticity [7,8]. This effect is positive when it leads to a hardening of the crystal under illumination and negative in the opposite case. The effect is believed to be due to the spectral properties of dislocations in semiconductors that are electrically active and sensitive to photonic excitation. The PPE has been extensively studied in AIIBVI semiconductors where generally a positive PPE has been reported. Relative strengthening up to 100% was observed in CdS [9]. The PPE of CdTe was studied in [10,11], and a positive PPE was found to be more pronounced in low resistivity samples.

Models explaining the PPE are usually based on an assumption of the interaction of dislocations with photosensitized centers [12]. The photoexcitation close to the band absorption edge results in an abundant creation of electron-hole pairs, which change the charge of both the dislocation line and point defects in the single crystal. Various types of differently charged centers may induce both positive and negative PPE. The appearance of the positive PPE as a result of the illumination-induced increase in the Peierls barrier was suggested [13]. Recent density functional theory (DFT) calculations for zinc-blende ZnS [14] connect the positive PPE with the formation of energetically more favorable bonds at the dislocation cores due to trapping of the photogenerated free carriers and the necessity to break these bonds upon dislocation glide. The mobility of the dislocations is therefore reduced as a result of the modification of the distribution of the electric charge along the dislocation lines when compared to the situation without illumination. This results in the experimentally observed positive PPE.

Dislocations are also of high interest in CdTe and related compounds (CdZnTe, CdZnTeSe), because they form defect states within the bandgap and influence charge trapping and recombination in devices like X-ray and gamma-ray detectors [15,16,17] and solar cells [18,19].

The DFT calculations [20] show that the cores of several types of dislocation in CdTe may undergo lattice reconstructions along the dislocation lines accompanied with charge modulations resulting in the transformation of the originally metallic character to the semiconducting character of the reconstructed cores.

## 2. Materials and Methods

We studied two high resistivity detector-grade samples–Cd_1−x_Zn_x_Te (x = 0.08–0.10), further called CZT)–and a Cd_1−x_Zn_x_Te_1−y_Se_y_ (x = 0.08 (nominal x used was 0.1), y = 0.04, further called CZTS). Both crystals were prepared by the THM method. Measurements of the Pockels effect on CZT were performed on a neighboring sample that was used for microhardness measurements.

The samples were polished with final grit 0.05 μm before indentation. We characterized the mechanical properties of the material using a Qness Q10A automatic microhardness tester (Golling, Austria) with a Vickers load of 25 g and a test force time of 10 s. The evaluated microhardness values are denoted as HV0.025. We used the pyramidal Vickers-type (interfacial angle 136°) indenter, which produces a square impression. The microhardness value is calculated according to the formula HV = 1854.4 *P*/*d*^2^. In this equation, which has the dimension of stress, the load *P* and diagonal length *d* are measured in gram force and micrometer, respectively. For the load of 25 g force, the value of HV 0.025 is, therefore, HV0.025 = 46360/*d*^2^. To measure the photoplastic effect, we illuminated the spot below the indentation tip by light generated by a supercontinuum NKT SuperK Compact laser (NKT Photonics, Birkerød, Denmark) passing through narrow bandpass filters (Thorlabs, Newton, NJ, USA) with the central wavelength ranging between 750 and 1540 nm with steps of approximately 50 nm and full width at half maximum (FWHMs) between 10 and 40 nm. We also characterized it by room-temperature photoconductivity measurements with the same source that was used for the indentation experiment.

We measured the profile of steady-state electric field dependence on the incident near-infrared light photon energy at room temperature (300 K) by a cross-polarizers technique exploiting the Pockels effect of studied crystals [21,22,23]. The samples of dimensions of 5 × 5 × 2 mm^3^ were cut from the same single crystals as those used for microhardness studies. The sides were optically polished and the whole large opposite areas were equipped with gold and indium by evaporation forming the cathode and anode, respectively. The sample was biased to 500 V. Weak intensity testing light at a wavelength of 1550 nm passed through the optically polished sample placed between two crossed polarizers. The transmitted light was collected by the InGaAs camera. The map of the internal electric field was evaluated from the distribution of transmissivity.

## 3. Results

Initially, we measured the Vickers microhardness without illumination on the CZT sample and obtained the result HV0.025 = 74. Next, we repeated the measurement with illumination. The results are presented in Figure 1. It is apparent that the hardness increased at all illumination wavelengths by 4%–9%, indicating a positive photoplastic effect (PPE). At the longest wavelength of 1550 nm (0.8 eV), which corresponds to approximately half the bandgap of the studied CZT sample at room temperature, the observed effect was the smallest. With a decreasing wavelength down to 1400 nm (~0.88 eV), the PPE slightly increases. This part of the measuring dependence is followed by a small drop and a rather constant profile down to 1100 nm (~1.12 eV). Then, the hardness under illumination continues to increase until the last wavelength used in this study at 800 nm (1.55 eV), which corresponds to the bandgap of the sample at room temperature. The pivotal wavelengths at which we observed a change in the behavior of the dependence are, therefore, at ~1400 nm and ~1100 nm.

The results of the photoconductivity measurements are presented in Figure 2. The signal increases with decreasing wavelength for below band-gap radiation. We observed a small maximum at 1100 nm.

Figure 3 shows a correlation diagram between the Vickers hardness HV0.025 (Figure 1) and photoconductivity (Figure 2). In general, we observe a good correlation between both physical quantities. The dependence is characterized by three spectral regions separated at wavelengths of 1100 nm and 1400 nm. Within each region the HV0.025 increases with the decreasing wavelength of the illuminating light.

We repeated the set of indentation and photoconductivity measurements on the CZTS sample. The results of the indentation measurements with additional illumination are presented in Figure 4.

The HV0.025 increases with decreasing wavelength of the illuminating light. There is a sharp increase at about 1250 nm (~1 eV); the increase is rather small until approximately 1000 nm (~1.24 eV) at which point a new increase in hardness can be observed.

The spectral dependence of the photoconductivity of the CZTS sample is shown in Figure 5. It is characterized by a gradual increase in the signal with decreasing wavelength of the light up to the bandgap energy when the signal starts to decrease. We can observe that at wavelengths of ~1200 nm (1.03 eV) and 1100 nm (1.12 eV), the increase in the photoconductivity signal is accelerated. The correlation diagram between photoconductivity and HV0.025 for the CZTS sample is presented in Figure 6. Both physical quantities are well correlated. There is an observable change of the slope of the dependence at ~1200 nm, reflecting the increase of HV0.025 at the same wavelength (Figure 4).

Based on these results we assume, therefore, that the spectral dependence of the photoinduced changes of HV0.025 (Figure 1) is connected to illumination-generated transitions of the electrons from deep levels to the conduction or valence bands at energies of ~0.88 eV (1400 nm) and ~1.1 eV (1100 nm) in CZT and at ~1 eV (1250 nm) in CZTS.

To clarify whether these transitions are to the conduction or valence bands, we measured the spectral dependence of the relative change of the local space charge below the cathode upon illumination of both samples by the Pockels electro-optic effect (Figure 7). We can see that in the case of the CZT sample the relative local space charge density is positive. The positive space charge starts to increase strongly at 1000 nm (1.24 eV). Electrons are transferred by light from the deep level to the conduction band leaving an uncompensated positive space charge. At ~1400 nm (~0.88 eV), the change of the signal is too small to make a clear conclusion.

The response of the CZTS sample to below band-gap illumination was much weaker compared to the CZT sample, probably due to smaller concentrations of deep levels responsible for charge trapping. We observed an increase in the local space charge density starting at ~1000 nm, which correlates with an increase of HV0.025 (Figure 4).

The relation of the E_C_-1.1 eV deep level to dislocations in CdMnTe was shown in ref. [24] using I-DLTS measurements to study the filling dynamics of the deep trap. The observed logarithmic dependence corresponds to plastically deformed Si and GaAs materials [25,26].

The results of the indentation and photoconductivity measurements on both the CZT and CZTS samples can be summarized as follows:The microhardness HV0.025 increases upon illumination over the whole applied spectral range (1540–750 nm), indicating a hardening of the material.The photoconductivity and Vickers microhardness HV0.025 are well correlated. At specific wavelengths we observed changes in the correlation diagram (change of slope and an abrupt decrease of HV0.025) with decreasing wavelength of the illuminating light.

The deep level at ~E_C_-1.1 eV strongly influences the PPE. This indicates that the change in PPE is related to dislocations, which is consistent with previous observations from I-DLTS measurements in CdMnTe that found the 1.1 eV level was related to dislocations.

## Figures and Tables

**Figure 1 materials-14-01465-f001:**
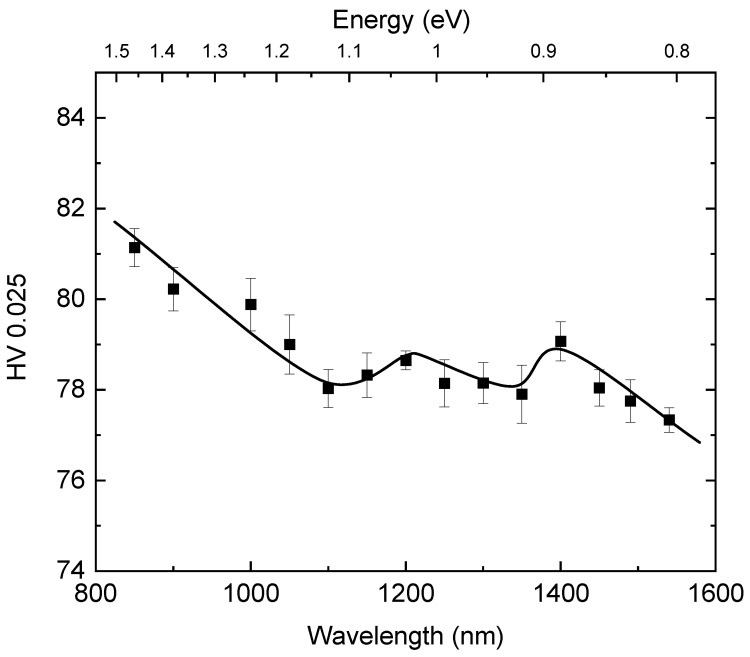
Evolution of Vickers microhardness HV0.025 on the wavelength of the illuminating light measured on a CZT sample.

**Figure 2 materials-14-01465-f002:**
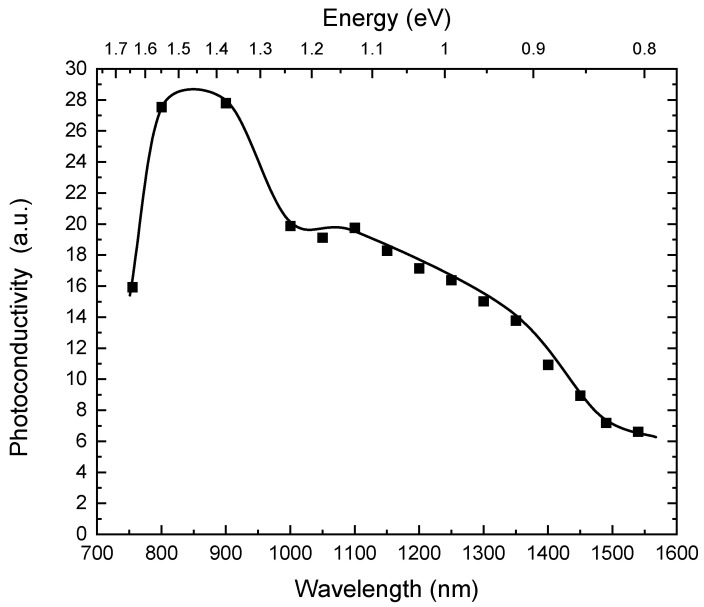
Spectral dependence of the photoconductivity for a CZT sample.

**Figure 3 materials-14-01465-f003:**
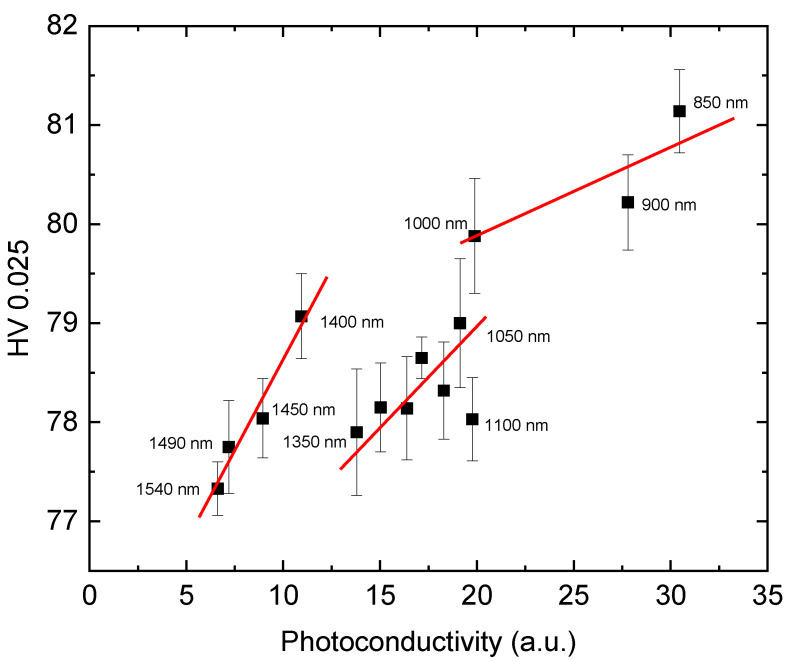
Correlation diagram between the Vickers microhardness HV0.025 and photoconductivity (CZT sample).

**Figure 4 materials-14-01465-f004:**
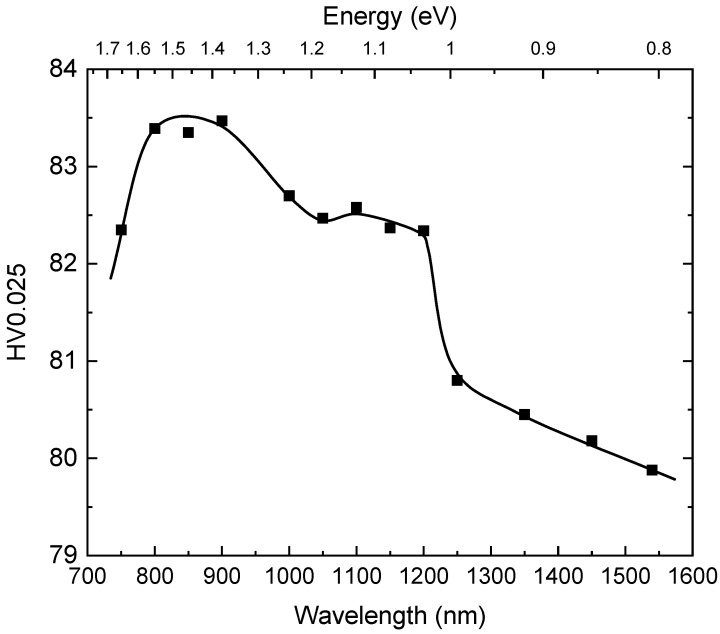
Vickers microhardness as a function of the illuminating light for CZTS.

**Figure 5 materials-14-01465-f005:**
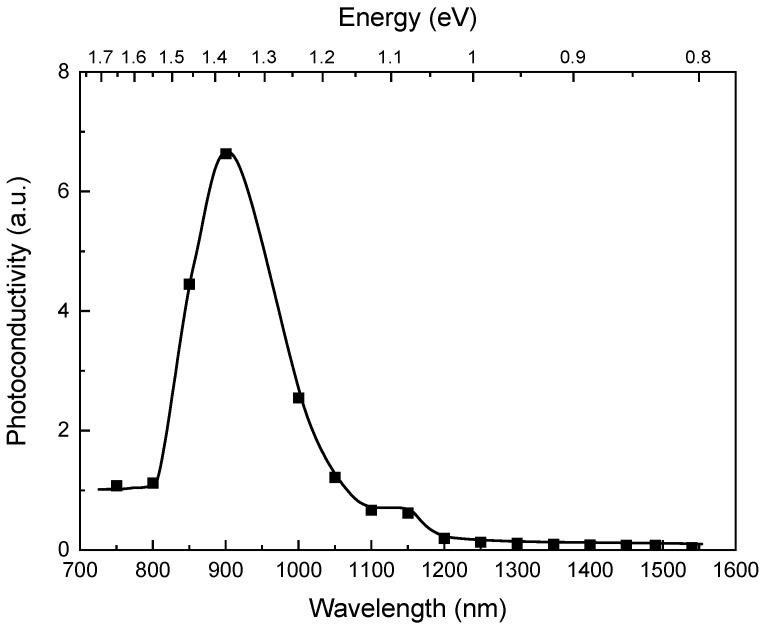
Spectral dependence of the photoconductivity for CZTS.

**Figure 6 materials-14-01465-f006:**
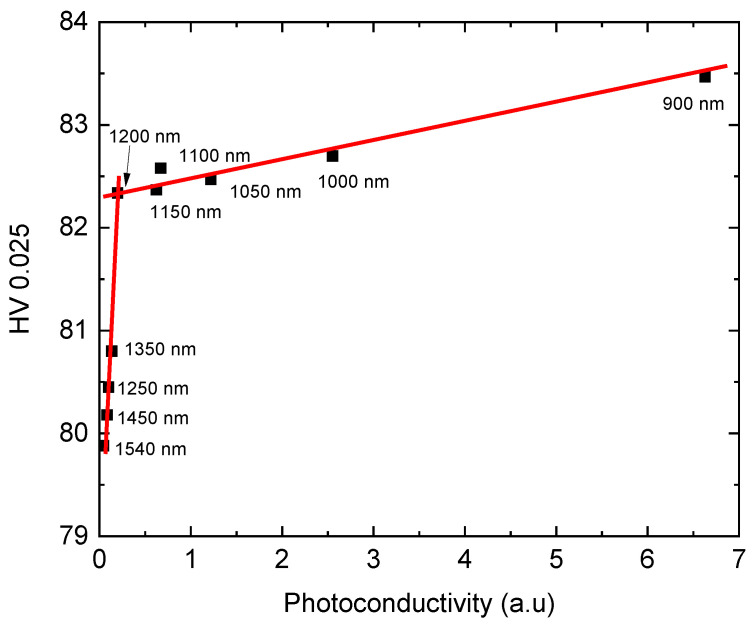
Correlation diagram between the Vickers microhardness HV0.025 and photoconductivity for a CZTS sample.

**Figure 7 materials-14-01465-f007:**
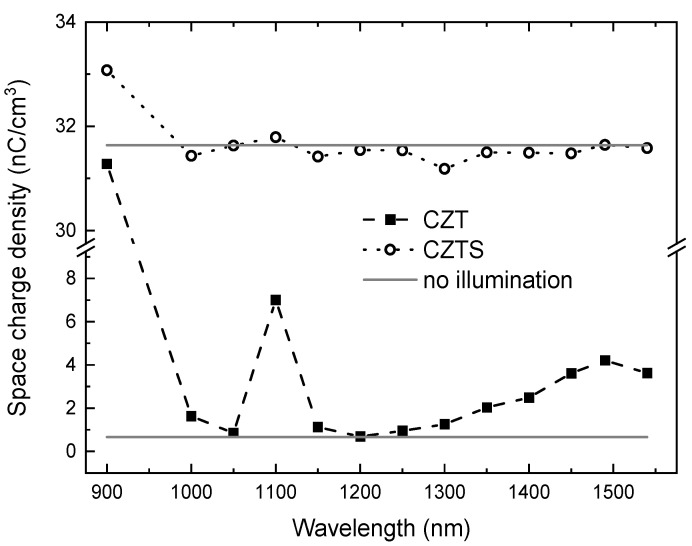
Spectral dependence of the relative change of internal space charge for the CZT and CZTS samples.

## Data Availability

The data presented in this study are available on request from the corresponding author.

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
