# Peer review of "Spectral Dependence of the Photoplastic Effect in CdZnTe and CdZnTeSe"

_materials, 2021, doi:10.3390/ma14061465_

Round 1
Reviewer 1 Report
In this paper the authors describe spectral dependence of photo plastic effect in CZT and CZTSe. It is a well written paper and add value to radiation detection based on CZT. I therefore recommend it to be published with a few minor amendments and a possible further proof read for mistakes.
-Please add a description of Vickers microhardness HV 0.025
-Please add the shape of the electrode configuration. There is some dependencies were observed based on the size of the electrode in"Wang, Y., Tao, L., Abbaszadeh, S., & Levin, C. S. (2021). Further investigations of a radiation detector based on ionization-induced modulation of optical polarization. Physics in Medicine & Biology". You might consider mentioning this in the discussion.
Author Response
We thank the reviewers for the careful assessment of the revised manuscript. We dealt with and addressed all remarks and recommendations. We believe that the revised manuscript was improved considerably by the amendments. The reviewers' comments are highlighted blue, the comments are black.
Reviewer #1: In this paper, the authors describe the spectral dependence of the photo plastic effect in CZT and CZTSe. It is a well-written paper and adds value to radiation detection based on CZT. I, therefore, recommend it to be published with a few minor amendments and a possible further proofread for mistakes.
-Please add a description of Vickers microhardness HV 0.025 .
We added the following paragraph to the section Materials and Methods:
line 75: We used the pyramidal Vickers-type (interfacial angle 136°) indenter, which produces a square impression. The microhardness value is calculated according to the formula HV=1854.4 P/d2. In this equation, which has the dimension of stress, the load P and diagonal length d are measured in gram force and micrometer, respectively. For the load of 25 gram force, the value of HV 0.025 therefore is HV 0.025 = 46360/d2
-Please add the shape of the electrode configuration. There is some dependencies were observed based on the size of the electrode in"Wang, Y., Tao, L., Abbaszadeh, S., & Levin, C. S. (2021). Further investigations of a radiation detector based on ionization-induced modulation of optical polarization. Physics in Medicine & Biology". You might consider mentioning this in the discussion.
Response: The effect of electrode shapes on the electric field distribution is a very well-known phenomenon. The electrodes were covering whole areas of large opposite sides of the samples with dimensions of 5x5x2 mm. Therefore, the electric field was evaluated from the central part of the sample. We agree that the sample configuration for Pockels measurements is described very briefly in the original manuscript. Therefore we have added the following text to the section Materials and Methods:
The samples of dimensions of 5x5x2 mm3 were cut from the same single crystals as those used for microhardness studies. The sides were optically polished and the whole large opposite areas were equipped with gold and indium by evaporation forming the cathode and anode, respectively. The sample was biased to 500 V.
line 87: ...from the distribution of transmittance from the central part of the sample.
We have also added the citation you recommended and a citation of our recent paper dealing with the more general shape of electrodes.
Reviewer 2 Report
Review of "Spectral dependence of the Photo plastic effect in CdZnTe and
CdZnTeSe" by Franc, Dědič, Moravec, Rejhon, Grill, Elhadidy, Vladimír , Šíma , Cieslar, Bratko, Roy, and James
This paper investigates the spectral dependence of the Vickers Hardness of two semiconductor compounds CdZnTe (CZT) and CdZnTeSe (CZTS). The authors filter out narrow spectrum of light at wavelengths ranging from 800 nm to 1600 nm from a super-continuum source to illuminate onto samples of both CZT and CZTS. The authors then measure the micro Vickers hardness of the samples. The authors find at least a 9% increase in the hardness . the authors find a correlation between the increase in the hardness and the detected photocurrent. the authors measure the charge at the cathode and conclude that the generated photocarriers are associated with creation of more dislocations that increase the hardness of the compounds.
The paper is well-written and the material is very well presented, and I recommend that the paper be published as is.
I do have the following minor questions:
Does the hardness of the material revert back to the original hardness when illumination is stopped?
Is there a different sample used for each illumination by each different wavelength?
The error bars in Figures 1 and 3, were there many independent measurements taken at at the various wavelengths ?
Author Response
We thank the reviewers for the careful assessment of the revised manuscript. We dealt with and addressed all remarks and recommendations. We believe that the revised manuscript was improved considerably by the amendments. The reviewers' comments are highlighted blue, the comments are black.
Response to reviewer 2
Does the hardness of the material revert back to the original hardness when illumination is stopped?
Yes, the microhardness always reverted to its original value in the dark. During our experiments, we regularly checked this value to rule out possible instabilities in the crystal.
Is there a different sample used for each illumination by each different wavelength?
The maximum dimensions of our samples of about 15x12 mm2 allowed us to measure the entire spectral dependence on a single sample. The individual indentation punctures had a diagonal up to 30 μm and the distance between them was 200 μm.
Reviewer 3 Report
The authors here investigate spectral dependence of the Vickers microhardness HV 0.025 of CdZnTe and CdZnTeSe samples upon illumination. They found that the photoconductivity and microhardness are correlated. The data is convincing and the discussion is based on the experiments results. However, the following should be addressed.
- From Figure 1, the author declared that the signal increases with decreasing wavelength for below band-gap radiation. What is the exact bandgap of CZT used in this experiment? Can the author presente the absorbance of this CZT sample?
- Figure 3, shows a correlation diagram between the Vickers hardness HV 0.025 and photoconductivity. The author declared that within each region the HV0.025 increases with decreasing wavelength of the illuminating light. However, it is evident that the HV0.025 first increases with are decreasing wavelength then drop down when further decrease the wavelength. The author should explain this more detail.
Author Response
We thank the reviewers for the careful assessment of the revised manuscript. We dealt with and addressed all remarks and recommendations. We believe that the revised manuscript was improved considerably by the amendments. The reviewers' comments are highlighted blue, the comments are black.
Reviewer #3: The authors here investigate spectral dependence of the Vickers microhardness HV 0.025 of CdZnTe and CdZnTeSe samples upon illumination. They found that the photoconductivity and microhardness are correlated. The data is convincing and the discussion is based on the experiment results. However, the following should be addressed.
- From Figure 1, the author declared that the signal increases with decreasing wavelength for below band-gap radiation. What is the exact bandgap of CZT used in this experiment? Can the author present the absorbance of this CZT sample?
We estimate the bandgap to be 1.59eV. The light used for illumination is well below bandgap. We did not measure exact absorbance of the sample.
- Figure 3, shows a correlation diagram between the Vickers hardness HV 0.025 and photoconductivity. The author declared that within each region the HV0.025 increases with decreasing wavelength of the illuminating light. However, it is evident that the HV0.025 first increases with are decreasing wavelength then drop down when further decrease wavelength. The author should explain this in more detail.
We have added the following more elaborate explanation in the text.
line 130: The relative drop of HV 0.025 in the wavelength interval 1100 nm – 1400 nm (1.1 eV – 0.9 eV) may be consistently interpreted by considering a defect model of CdZnTe with deep levels identified by the Photo-Hall Effect Spectroscopy [A. Musiienko, R. Grill, P. Hlídek, P. Moravec, E. Belas, J. Zázvorka, G. Korcsmáros, J. Franc and I. Vasylchenko. Deep levels in high resistive CdTe and CdZnTe explored by photo-Hall effect and photoluminescence spectroscopy, Semicond. Sci. Technol. 2017, 32 015002]. While at the low-energy near 0.8 eV and at the high-energy above 1.1 eV the optical transitions from the deep level-to-conduction band dominate, the intermediate interval 0.9 eV - 1.1 eV comprises mainly valence band-to-deep level antipodes. We deduce that the optical transitions representing electron photoexcitation from deep levels to the conduction band inspire the lattice hardening while oppositely valence band-to-deep level transitions induce the softening. This feature seems to us logical since the excess of holes disrupts the covalent bonds in the crystal decreasing accordingly its stability.